# Exploring the Impact of Chemotherapy on the Emergence of Antibiotic Resistance in the Gut Microbiota of Colorectal Cancer Patients

**DOI:** 10.3390/antibiotics14030264

**Published:** 2025-03-05

**Authors:** Mutebi John Kenneth, Chin-Chia Wu, Chuan-Yin Fang, Tsui-Kang Hsu, I-Ching Lin, Shih-Wei Huang, Yi-Chou Chiu, Bing-Mu Hsu

**Affiliations:** 1Department of Earth and Environmental Sciences, National Chung Cheng University, Chiayi 621, Taiwan; 2Doctoral Program in Science, Technology, Environment and Mathematics, National Chung Cheng University, Chiayi 621, Taiwan; 3Division of Colorectal Surgery, Dalin Tzu Chi Hospital, Buddhist Tzu Chi Medical Foundation, Chiayi 622, Taiwan; 4College of Medicine, Tzu Chi University, Hualien 970, Taiwan; 5School of Post-Baccalaureate Chinese Medicine, Tzu Chi University, Hualien 970, Taiwan; 6Division of Colon and Rectal Surgery, Ditmanson Medical Foundation Chia-Yi Christian Hospital, Chiayi 600, Taiwan; 7Department of Ophthalmology, Cheng Hsin General Hospital, Taipei 112, Taiwan; 8School of Medicine, National Yang Ming Chiao Tung University, Hsinchu 300, Taiwan; 9Department of Family Medicine, Asia University Hospital, Taichung 413, Taiwan; 10Department of Kinesiology, Health and Leisure, Chienkuo Technology University, Changhua 500, Taiwan; 11Center for Environmental Toxin and Emerging Contaminant Research, Cheng Shiu University, Kaohsiung 833, Taiwan; 12General Surgery, Surgical Department, Cheng Hsin General Hospital, Taipei 112, Taiwan

**Keywords:** chemotherapy, antibiotic resistance, SOS response, bloodstream infections, gut microbiota

## Abstract

With nearly half of colorectal cancer (CRC) patients diagnosed at advanced stages where surgery alone is insufficient, chemotherapy remains a cornerstone for this cancer treatment. To prevent infections and improve outcomes, antibiotics are often co-administered. However, chemotherapeutic interactions with the gut microbiota cause significant non-selective toxicity, affecting not only tumor and normal epithelial cells but also the gut microbiota. This toxicity triggers the bacterial SOS response and loss of microbial diversity, leading to bacterial mutations and dysbiosis. Consequently, pathogenic overgrowth and systemic infections increase, necessitating broad-spectrum antibiotics intervention. This review underscores how prolonged antibiotic use during chemotherapy, combined with chemotherapy-induced bacterial mutations, creates selective pressures that drive de novo antimicrobial resistance (AMR), allowing resistant bacteria to dominate the gut. This compromises the treatment efficacy and elevates the mortality risk. Restoring gut microbial diversity may mitigate chemotherapy-induced toxicity and improve therapeutic outcomes, and emerging strategies, such as fecal microbiota transplantation (FMT), probiotics, and prebiotics, show considerable promise. Given the global threat posed by antibiotic resistance to cancer treatment, prioritizing antimicrobial stewardship is essential for optimizing antibiotic use and preventing resistance in CRC patients undergoing chemotherapy. Future research should aim to minimize chemotherapy’s impact on the gut microbiota and develop targeted interventions to restore microbial diversity affected during chemotherapy.

## 1. Introduction

Colorectal cancer (CRC) remains a significant global health burden, ranking as the third most commonly diagnosed cancer and the second leading cause of cancer-related deaths worldwide [1]. Recent estimates indicate that 1.9 million new CRC cases occurred globally in 2022, while projections suggest a further increase in CRC incidence to 2.5 million new cases annually by 2035 [2]. The pathogenesis of CRC is driven by a complex interplay of genetic and environmental factors, and Western lifestyles and physical inactivity, and dietary habits, especially among the younger population [3,4]. However, early detection and timely intervention are crucial for improving patient outcomes.

While surgical resection remains the primary curative treatment [5], chemotherapy is a widely employed CRC therapeutic regimen, especially in advanced stages of the disease [6]. During this treatment, anticancer drugs are administered to target and destroy cancer cells, inhibit their proliferation, and prevent tumor invasion, and metastasis [7], with the ultimate goal of eradicating the disease. However, the adverse side effects associated with chemotherapy have spurred interests in developing less toxic treatment alternatives [8,9], including modulating the gut microbiota.

The gut microbiota is a complex ecosystem of microorganisms residing in the human gastrointestinal tract with a crucial role of maintaining a proper digestion and host immunity [10,11]. Emerging evidence suggests a strong link between gut dysbiosis and the development and progression of CRC [12,13,14]. This growing evidence has opened up new therapeutic avenues such as fecal microbiota transplantation (FMT), probiotics, and prebiotics, which aim at restoring the altered gut microbiota to improve patient survival and the efficacy of CRC therapeutic regimens [15,16,17,18,19,20].

However, certain anticancer regimens may lead to the development of antimicrobial resistance (AMR) in the gut microbiota, increasing the risk of antibiotic-resistance-associated complications and CRC-related mortality [21,22,23]. Additionally, these regimens, particularly chemotherapy, not only disrupt gut microbiota homeostasis but also induce dysbiosis [24,25]. The induced dysbiosis creates a conducive environment for the proliferation of antibiotic-resistant bacteria [26,27]. Moreover, chemotherapy can activate the bacterial SOS response and accelerate bacterial mutations and horizontal gene transfer, thus driving the emergence of antibiotic-resistant mutants within the gut ecosystem [23,28,29]. Furthermore, chemotherapy compromises the integrity of the intestinal barrier, increasing the susceptibility to infections [30], and recurrent septic episodes [31]. This infection vulnerability often requires continued antimicrobial therapy, which may inadvertently contribute to the development of AMR pathogens [32,33]. These facts suggest that chemotherapy may increase the risk of superbug infections in CRC patients, potentially hindering their recovery and raising mortality rates. In this review, we explore how the prolonged use of antibiotics during chemotherapy, combined with chemotherapy-induced bacterial mutations, can intensify antibiotic resistance in CRC patients, ultimately contributing to poor treatment outcomes.

## 2. Chemotherapy for Colorectal Cancer Treatment

The five-year survival rate for patients diagnosed with early-stage CRC exceeds 60% [34]. However, over 50% of CRC patients are diagnosed at advanced stages [35], often with distant metastases, reducing survival rates to approximately 10% [36]. In such cases, surgical resection is rarely curative, necessitating chemotherapeutic intervention.

Chemotherapy is a cornerstone CRC treatment [37], used in adjuvant and palliative settings, especially in advanced or metastatic CRC (mCRC) [38,39]. Its application depends on the cancer’s stage, molecular characteristics, and the patient’s overall health [40]. This form of cancer treatment involves the systemic administration of anticancer agents that target to kill or inhibit rapidly dividing cancer cells.

Chemotherapy is categorized into two main approaches for CRC. Neoadjuvant chemotherapy, administered before surgery, is primarily used in locally advanced CRC to reduce tumor burden and improve resectability [41]. In contrast, adjuvant chemotherapy, given postoperatively, aims to eradicate micro-metastases and lower the risk of recurrence [38,42].

Therapeutic agents like fluoropyrimidines—including 5-fluorouracil (5-FU) and capecitabine—form the backbone of CRC chemotherapy [43]. These are often combined with oxaliplatin or irinotecan in regimens such as FOLFOX, CAPOX, and FOLFIRI to enhance efficacy and improve survival rates [44].

Despite their potent cytotoxic effects against cancer cells, these agents are associated with significant adverse effects that impair patients’ quality of life [9,45]. Hematological toxicities, such as neutropenia and anemia, gastrointestinal symptoms including diarrhea and mucositis, and neurotoxicity associated with oxaliplatin are often experienced by patients undergoing chemotherapy [46]. These often lead to dose adjustments and treatment delays, potentially compromising therapeutic efficacy [47,48].

Beyond these systemic effects, chemotherapy profoundly affects the gut microbiota [15,20,24]. The gut microbiota plays a crucial role in modulating the efficacy and toxicity of chemotherapeutic agents [20,49]. However, chemotherapy-induced mucositis disrupts the gut barrier and creates an inflammatory environment [50,51], altering the microbial composition and diversity [24]. This disruption contributes to secondary infections, thus complicating CRC treatment outcomes even further [52,53]. The interplay between the gut microbiota and chemotherapeutic drugs as illustrated in Figure 1 indicates how modulating the gut microbiota during chemotherapy could improve treatment outcomes.

The increased vulnerability to infections implies prolonged antimicrobial therapy, heightening the risk of AMR and additional threats to patient survival [54,55,56]. Given the central role of chemotherapy in advanced CRC treatment, implementing gut-microbiota-targeted interventions is crucial for preventing dysbiosis, mitigating AMR, and improving treatment outcomes.

## 3. Gut Microbial Modulation of the Efficacy and Toxicity of Chemotherapy in Colorectal Cancer

Chemotherapy is a widely recommended treatment for advanced CRC; however, patient responses to this therapy exhibit considerable variability [57]. A significant factor contributing to this variability could be the composition of the patient gut microbiota [20,58,59]. A summary of the influence of gut bacteria in chemotherapy toxicity and efficacy is illustrated in Table 1. Specifically, recent studies indicate that the gut microbiota can modulate the efficacy and toxicity of chemotherapy through mechanisms outlined in the “TIMER” framework proposed by Alexander et al.: translocation, immunomodulation, metabolism, enzymatic degradation, and reduced diversity and ecological variation [60,61].

**Table 1 antibiotics-14-00264-t001:** Summary of gut microbiota modulating the efficacy and toxicity of chemotherapeutic drugs.

Therapy	Model/Assay	Gut Microbiota Involved	Mechanism of Interaction	Reference
Irinotecan (SN38)	Mice	β-glucuronidase-producing gut bacteria	Chemical transformation of inactive SN38G into active SN38, inducing severe toxic effects	[62]
CB1954 (pro-drug of gemcitabine)	In vitro	*Escherichia coli*	Bacterial nitroreductase activity amplifies CB1954 activity	[63]
Gemcitabine	Mice	γ-proteobacteria, *Mycoplasma hyorhinis*, and *Escherichia coli*	Inactivation of gemcitabine to its inactive form 2′,2′-difluorodeoxyuridine by bacteria Cytidine deaminase	[64]
5-FU	*C. elegan*	*Escherichia coli*	Interconversion of the 5-FU with vitamin B6 and B9 release and ribonucleotide metabolism	[65]
	Mice and in vivo	*Fusobacterium nucleatum*	Causes chemoresistance by activating the autophagy pathway	[66]
	In vitro and mice	Gut microbiota	Causes gut dysbiosis and mucositis	[67,68]
	Mice	*Bacteroides vulgatus*	*Bacteroides vulgatus*-mediated nucleotide biosynthesis induces 5-fluorouracil resistance	[69]
Methotrexate	In vitro	*Escherichia coli*	The drug selects for antibiotic resistance	[70]
Oxaliplatin	Mice and in vivo	*Fusobacterium nucleatum*	Causes chemoresistance by activating the autophagy pathway	[66]
Anti-PD-L1	Mice	*Commensal Bifidobacterium*	Promotes the antitumor immunity and facilitates anti-PD-L1 efficacy	[71]
Etoposide	In vitro	*Pseudomonas aeruginosa*	Oxidative stress drives the emergence of antibiotic resistance to fluoroquinolones	[72]
Doxorubicin	Mice	*Akkermansia muciniphila*	Enhanced the responsiveness of doxorubicin in triple-negative breast cancer (mice model)	[73]

In the context of the “TIMER” framework, one critical aspect in chemotherapy modulation is that chemotherapy-induced mucositis can compromise gut epithelial integrity, facilitating microbial translocation into the bloodstream or lymphatic system [74]. This translocation may trigger systemic inflammation, sepsis, and immune dysregulation, ultimately diminishing the therapeutic outcomes of chemotherapy [52,75].

Still in the same framework, certain gut microbial metabolites also play a role in influencing chemotherapy outcomes [76,77,78]. For instance, butyrate—a short-chain fatty acid (SCFA) produced via dietary fiber fermentation by gut microbes—has been demonstrated to enhance the anticancer effects of oxaliplatin by modulating CD8+ T-cell function within the tumor microenvironment (TME) through IL-12 signaling [79,80]. Additionally, commensal microbes can impact the anticancer response to oxaliplatin by altering the activity of myeloid-derived cells in the TME [20,81]. This underscores the potential of gut microbial metabolites to enhance chemotherapy efficacy through immunomodulation [82].

Besides the above mechanisms, gut microbial enzymatic functions have been shown to play a pivotal role in influencing the metabolism and toxicity of chemotherapeutic agents [49]. For example, *Escherichia coli* has been shown to activate tegafur, a prodrug of 5-FU, and CB1954 through nitroreductases [63]. Conversely, *E. coli* inhibits drugs like vidarabine, cladribine, doxorubicin, and others, while enhancing the anticancer activity of mercaptopurine [49,62,63]. This further underscores how the mechanisms of the “TIMER” framework modulate the efficacy of chemotherapy through gut microbiota.

A notable toxicity interaction of the gut microbiota with chemotherapeutic drugs involves irinotecan, where bacterial β-glucuronidases convert the inactive form (SN38G) into the active form (SN38), leading to dose-limiting side effects such as irinotecan-induced diarrhea [62,83,84]. Given that β-glucuronidases are prevalent in various gut microbiota [85,86], the co-administration of β-glucuronidase inhibitors during irinotecan therapy has been shown to mitigate diarrhea. This highlights the potential for the strategic modulation of gut microbiota to enhance chemotherapy efficacy while minimizing toxicity.

Certain gut bacteria have been implicated in promoting chemotherapy resistance, thereby undermining treatment outcomes [87]. For example, *Fusobacterium nucleatum*, which is enriched in CRC patients, has been associated with worse clinical outcomes, including chemoresistance to 5-FU and oxaliplatin through autophagy modulation [66,88]. Furthermore, *F. nucleatum* has been reported to upregulate BIRC3, which directly inhibits the caspase cascade, thus leading to 5-FU chemoresistance [89]. This evidence suggests that the targeted modulation of the gut microbiota against opportunistic microbes like *F. nucleatum* present promising strategies for reducing chemoresistance and improving therapeutic outcomes for patients with advanced CRC.

## 4. Chemotherapy-Induced Gut Dysbiosis in Colorectal Cancer

Recent studies highlight the crucial role of the gut microbiota in modulating the efficacy and toxicity of chemotherapeutic agents, making it a promising target for improving drug safety and efficacy through microbiota manipulation [49,90,91,92]. However, chemotherapy has been shown to induce significant perturbations in the gut microbiota, with pre- and post-treatment fecal analyses across various cancer cohorts as summarized in Table 2, which highlights its contribution to dysbiosis [24,25,93,94,95]. For instance, a recent study reported a significant reduction in gut microbiota in breast cancer patients undergoing chemotherapy [96]. Similarly, a study on lung cancer patients found that pemetrexed disrupted the gut microbial balance and compromised the colon barrier integrity [97]. In patients with non-Hodgkin’s lymphoma, chemotherapy was also associated with profound disruptions in the intestinal microbiome, affecting both its taxonomic composition and metabolic capacity [93]. These studies consistently indicate that chemotherapy depletes beneficial gut bacteria, while promoting the growth of opportunistic pathogens. This suggests that, while chemotherapy is the cornerstone of advanced CRC treatment, its side effects—including disrupting and depleting beneficial gut flora—can significantly impact patient recovery and quality of life. The depletion of such flora is particularly concerning as the gut microbiota modulates chemotherapy efficacy and mitigates chemotoxicity [49].

Adjuvant chemotherapy is commonly administered in advanced CRC patients to minimize recurrence [98,99]. However, its cytotoxic effects to normal cells and gut microbiota contribute significantly to side effects that impair one’s quality of life [100,101]. For example, chemotherapy-induced mucositis, driven by reactive oxygen species (ROS) formation [50] and the production of pro-inflammatory cytokines such as interleukin-1β (IL-1β), IL-6, and tumor necrosis factor-α (TNF-α) [102], leaves mucosal tissue vulnerable to infection and ulceration. It is also reported that most chemotherapeutic drugs harm not only normal epithelial cells but also commensal bacteria, highlighting their non-selective toxicity and contribution to dysbiosis [24].

The accumulating evidence links chemotherapy-induced mucositis as the cause of gut dysbiosis in patients undergoing chemotherapy [24,50,103,104]. Irinotecan, a key drug for metastatic CRC, exemplifies this by the reported damage caused to both epithelial cells and gut microbiota, leading to gastrointestinal toxicity and dysbiosis [105]. Similarly, 5-FU, a first-choice chemotherapeutic drug of CRC treatment, has been associated with mucositis and significant gut microbial alterations [104,106,107]. Preclinical studies have observed a shift from beneficial bacteria like *Lactobacillus* and *Bifidobacterium* to pathogenic genera such as *Clostridium*, *Escherichia*, and *Enterococcus* following a single dose of 5-FU [108]. These disruptions, as well as those summarized in Table 2, underscore the effect of chemotherapy on gut health and how this can be targeted to improve patient quality of life.

**Table 2 antibiotics-14-00264-t002:** Summary of how chemotherapy drugs affect gut microbial community in different cancers.

Cancer Type	Model/Assay	Chemotherapy Regimen	Effects on Gut Microbiota	Reference
Lung cancer	Mice	Pemetrexed	Decrease in bacterial family of SCFA-producing taxa *Ruminococcaeae*. A significant increase in two opportunistic bacterial families, *Enterobacteriaceae*, and *Enterococcaceae*	[97]
Stage III CRC	Human	Capecitabine-Oxaliplatin (CapeOx)	Predominance of opportunistic *Klebsiella pneumoniae* (31%) in patients with chemotherapy-induced diarrhea	[109]
CRC	Human	5-FU + oxaliplatin	Enrichment of *Veillonella dispar*, *Bacteroides plebeius*, and *Prevotella copri* in patients treated with this regimen	[110]
Breast, colorectal, esophageal, laryngeal, and melanoma	Human	Capecitabine, cisplatin/5-FU, FOLFOX4, FOLFOX6, FOLFIRI, 5-FU/folinic acid, paclitaxel, carboplatin, and gemcitabine	Reduced abundance of beneficial gut bacteria such as *Lactobacillus* spp., *Bacteroides* spp., *Bifidobacterium* spp., and *Enterococcus* spp., and increased abundance of opportunistic species such as *Staphylococcus* spp. and *Escherichia coli* were observed in patients undergoing chemotherapy	[111]
Ovarian cancer	Human	Paclitaxel, carboplatin, and cisplatin	Altered gut microbial composition in ovarian cancer patients, marked with increased abundance of *Bacteroidetes* and *Firmicutes* and a decreased abundance of *Proteobacteria* after chemotherapy	[112]
Breast cancer	Human	Taxane, cyclophosphamide, carboplatin, and doxorubicin	Altered gut microbiota which taxonomic shifts in *Fusicatenibacter*, *Faecalibacterium*, *Erysipelotrichaceae* UCG-003, *Bacteroides*, and *Subdoligranulum*, leading to cognitive decline during chemotherapy	[113]
CRC	Human	5-FU	Decreased abundance of *Deltaproteobacteria*, *Firmicutes*, and *Coriobacteria*; with increased mRNA levels of inflammatory cytokines in the gut such as TNF-α, IL-6, IL-1β, and IL-10 and nitric oxide synthase	[25]
Pancreatic ductal adenocarcinoma (PDAC)	Mice	Gemcitabine	The drug decreased the abundance of Gram-positive *Firmicutes* from about 39 to 17%, as well as the Gram-negative *Bacteroidetes* from 38 to 17%. However, the abundance of inflammation-associated bacteria such as *Akkermansia muciniphila* increased from 5 to 33%.	[114]

Recent studies have shown that the extent of chemotherapy-induced gut dysbiosis depends on the specific regimen administered. For example, the CapeOx regimen was associated with increased conditionally pathogenic bacteria, including *Bilophila*, *Anaerostipes*, *Comamonas*, *Collinsella*, *Bacteroides*, *Eggerthella*, and inflammation-related *Weissella* [109,115]. Another study which examined the effects of three regimens FOLFIRI, XELOX, and FOLFIRI combined with cetuximab showed varying impacts on microbial abundance. XELOX-treated patients exhibited an increase in the abundances of *Humicola*, *Tremellomycetes*, *Veillonella*, and *Malassezia*, alongside a decrease in the abundances of *Clostridiales*, *Faecalibacterium*, *Phascolarctobacterium*, *Rhodotorula*, and *Humicola*. Meanwhile, FOLFIRI altered the abundance of *Magnusiomyces*, *Candida*, *Dipodascaceae*, *Tremellomycetes*, and *Saccharomycetales*, whereas the abundances of *Rhodotorula* and *Humicola* were decreased [116]. In a combination therapy, the abundances of *Dipodascaceae*, *Candida*, *Magnusiomyces*, *Tremellomycetes*, *Saccharomycetales*, *Lentinula*, and *Malassezia* were increased in CRC patients treated with the FOLFIRI regimen plus cetuximab compared with those treated with the FOLFIRI regimen alone [116]. These findings emphasize that interventions targeting chemotherapy-induced gastrointestinal disorders or mucosal inflammation should consider regimen-specific microbial effects, as different regimens may exert distinct impacts on the gut microbiota.

## 5. Chemotherapy-Induced Microbial Mutations and the Emergence of De Novo AMR in Gut Microbiota

In addition to gut microbial perturbation, chemotherapeutic agents have been reported to induce mutations within the gut microbiota [117]. The mechanisms by which these agents operate often involve damaging bacterial DNA and inhibiting DNA replication, thereby exerting selective pressure that influences the evolution of gut bacteria [118]. This selective pressure can lead to increased genotypic diversity, proliferation, and adaptations among the bacterial populations selected for, within the gut ecosystem. Damaging bacterial DNA activates the SOS response, a cellular mechanism that drives de novo mutations [119]. First described by Radman in 1975, the SOS response is triggered by the accumulation of single-stranded DNA breaks, leading to the formation of a nucleoprotein complex with the ubiquitous protein RecA. This complex cleaves the LexA repressor, thereby removing the inhibition of the SOS response [120]. Under the conditions created by chemotherapeutic regimens, DNA repair during the SOS response relies on low-fidelity DNA polymerases, which consequently heightens mutation rates [121].

Most chemotherapeutic agents (including alkylating agents and platinum-based compounds) as summarized in Table 3 strongly induce the SOS response in bacteria [122]. This response is also activated by nucleotide analogs, nucleotide synthesis inhibitors, and topoisomerase inhibitors [123]. The exposure of the gut microbiota to these agents through chemotherapy is likely to enhance bacterial mutagenesis through the activation of the SOS response [21]. A recent study investigating the induction of the SOS response in commensal *E. coli* with 39 chemotherapeutic drugs at therapeutic concentrations demonstrated that ten of these drugs activated the SOS response. Notably, eight of these accelerated the mutations in commensal *E. coli* through SOS activation [28]. This shows that chemotherapeutic agents could expedite the evolution of the gut microbiota and facilitate the emergence of resistant mutants within commensal bacteria.

Mutations arising from the activation of the SOS response may significantly contribute to the rapid emergence of antibiotic resistance among cancer patients [124,125]. This has been reported by numerous studies which indicated the emergence of antibiotic-resistant pathogenic bacteria in patients, attributed to SOS activation by chemotherapy drugs [21,22,23,28]. When gut bacteria acquire antibiotic resistance, they gain the ability to survive the exposure to antibiotic therapies that would otherwise eliminate them or inhibit their growth [126], which threatens patient survival.

**Table 3 antibiotics-14-00264-t003:** A summary of mechanisms underlying chemotherapy-induced antimicrobial resistance across different drugs and bacteria.

Chemotherapy Drug	Model/Assay	Mechanism of Antibiotic Resistance Emergence	Targeted Bacteria	Targeted Antibiotics	Reference
Cisplatin and oxaliplatin	In vitro	Induction of SOS response	*Escherichia coli* MG1655	Rifampicin, ciprofloxacin	[127]
39 chemotherapeutic drugs	In vitro	SOS-response induction	*Escherichia coli* MG1655, *Enterobacter cloacae* ATCC 1304, *Pseudomonas aeruginosa* ATCC 27853, and *Staphylococcus aureus* ATCC 25923	Ciprofloxacin, cefotaxime, imipenem	[28]
Methotrexate	In vitro	Selection for acquired resistance and co-selection for genetically linked resistance	*Escherichia coli*, and *Klebsiella pneumoniae*	Trimethoprim	[70]
Mercaptopurine, cytarabine, azacitidine, dacarbazine, daunorubicin, mitoxantrone, and cyclophosphamide	In vitro	Genotoxic effect stress activated the SOS response, thus increasing bacterial mutation	*Klebsiella pneumoniae* carbapenemase (KPC)-producing *Enterobacteriaceae*	Ceftazidime-avibactam	[128]
Etoposide	In vitro	Oxidative stress	*Pseudomonas aeruginosa*	Ciprofloxacin	[72]
Paclitaxel and its derivative docetaxel	In vitro	Upregulation of rpoS expression, activated SOS response, conjugative transfer of resistance genes	*Escherichia coli*	Rifampicin and ampicillin	[129]

A recent study involving seven clinical isolates of *Enterobacteriaceae* producing KPC-type carbapenemases revealed that mutations induced by anticancer therapies in experimental bacteria increased the resistance against ceftazidime-avibactam and rifampicin up to 10^4^-fold compared to no resistance in the control experiment [128]. This phenomenon highlights an additional layer of complexity and threat that chemotherapy drugs create on the gut microbiota.

Besides the activation of the SOS response, recent studies have implicated chemotherapeutic drugs in inducing antibiotic resistance in tumor-associated bacteria through oxidative stress [22,128]. For example, a recent study stated that oxidative stress induced by Etoposide (ETO), an FDA-approved chemotherapy drug, drives the emergence of *Pseudomonas aeruginosa* resistance to fluoroquinolones [72]. However, another study indicated that a widely used chemotherapeutic drug methotrexate induces the selection of antibiotic resistance in *E. coli* [70], which implies that chemotherapy may promote the spread of antibiotic resistance genes. In addition, paclitaxel and its derivative docetaxel have been reported to promote the transfer of antibiotic resistance through a series of cellular responses such as an enhanced cell membrane permeability, activated SOS response, increased plasmid replication, pilus formation, and the upregulated expression of *rpoS* gene [129]. These studies indicate that chemotherapy can also induce selective pressure against the gut microbiota, encouraging them to acquire and transfer antibiotic resistance. This highlights the need for novel anti-cancer regimens with minimal effects on the gut microbiota.

Horizontal gene transfer (HGT) events are crucial for the rapid acquisition of complex antibiotic resistance mechanisms within bacterial ecosystems under selective pressure [129,130,131]. HGT can occur through bacteriophage-mediated transduction, transformation with extracellular DNA, or plasmid exchange via bacterial conjugation [132]. The rate of HGT—particularly through prophage induction—increases substantially during the SOS response, suggesting that a heightened rate of SOS activation due to chemotherapy may facilitate a rapid spread of antibiotic resistance in a patient under chemotherapy, posing additional threats to the patient health [133]. The combined effects of the SOS response on commensal microbiota and increased HGT rates resulting from chemotherapy can contribute to elevated rates of plasmid exchange within the gut microbiota. This further exacerbates bacterial adaptation and the dissemination of acquired mutations, including those conferring antibiotic resistance [133]. Therefore, the outlined risk of acquiring antibiotic resistance during chemotherapy calls for future research to focus on developing safer chemotherapeutic regimens and strategies to protect the gut microbiota against them.

## 6. Effects of Antibiotic–Chemotherapy Combinations on Persistence of AMR in Gut Microbiota

Bacterial infections are prevalent among CRC patients undergoing chemotherapy [134,135]. This is primarily due to mucositis, which creates open sores that facilitate bacterial colonization, and could also be due to the direct effects of chemotherapy that induce dysbiosis in the gut microbiota [50,103,136]. Consequently, these patients rely on effective antibiotics for both the prevention and treatment of infections [137]. Mucositis further heightens the risk of bacterial translocation into the bloodstream, necessitating additional antibiotic therapy [138,139]. This cycle can lead to increased selection pressure for antibiotic-resistant bacteria, resulting in their persistence within the gut microbiota. The predominance of resistant pathogens poses a significant risk to CRC patient survival, as bacterial infections represent one of the most frequent complications in cancer care [140,141,142]. The failure of antibiotic treatment could lead to a higher incidence of systemic infections and the associated mortality [142]. Moreover, cancer patients are particularly vulnerable to hospital-acquired infections; yet, hospitals are major reservoirs of antibiotic-resistant bacteria [143,144,145]. This vulnerability exacerbates the antibiotic resistance and increases the mortality associated with antibiotic failure. This explains why over 40% of oncologists in the United Kingdom express concern that rising antibiotic resistance threatens the effectiveness of cancer treatments [22].

Similar to chemotherapeutic agents, most antibiotics used to treat bacterial infections have been shown to activate the SOS response [124,125]. For instance, quinolones such as ciprofloxacin inhibit bacterial topoisomerases II and IV, leading to double-stranded DNA breaks that trigger the SOS response [146,147]. This suggests that combining antimicrobial prophylaxis with chemotherapy may inadvertently heighten the risk of de novo antibiotic-resistance mutations in gut bacteria. This reverses the intended mitigation of bacterial infections to potentially increasing mortality in CRC patients due to antibiotic failure—even though antibiotic-resistant infections are rarely recorded as direct causes of death on death certificates [22]. This aligns with recent comparative studies which indicate that cancer patients undergoing chemotherapy with antimicrobial prophylaxis have lower overall survival (OS) than those without prophylaxis [148,149,150,151]. For instance, a study on the impact of antibiotic treatment during platinum chemotherapy for epithelial ovarian cancer found that antibiotics reduced OS in the treatment group compared to the control group (45.6 vs. 62.4 months). Notably, using antibiotics that target Gram-positive bacteria further worsened OS, with a survival difference of approximately 17 months (35.0 vs. 62.4 months) [148]. Similarly, a study on non-small-cell lung cancer (NSCLC) patients revealed that antibiotics treatment one month prior to chemotherapy significantly decreased OS compared to those who did not receive antibiotics (13.8 vs. 17.6 months, *p* < 0.001) [151]. These findings indicate that antibiotic use before or during chemotherapy is consistently associated with poor treatment outcomes in cancer patients.

While modern cancer treatment approaches have significantly improved overall survival, they continue to render patients vulnerable to bacterial infections [142]. This implies that the emergence of antibiotic resistance could thus lead to unfavorable outcomes for patients who depend on antibiotics for infection prevention and treatment. A study by Bodro et al. [152] reported the increased persistence of bacteremia and higher early case-fatality rates among cancer patients infected with antibiotic-resistant ESKAPE pathogens (*Enterococcus faecium*, *Staphylococcus aureus*, *Klebsiella pneumoniae*, *Acinetobacter baumannii*, *P. aeruginosa*, and *Enterobacter* spp.) [152,153]. Additionally, another study found that 88% of deaths from hospital-acquired infections in an oncology intensive care unit were attributed to multidrug-resistant pathogens, underscoring the devastating impact of antibiotic resistance in cancer care [154].

In CRC care, chemotherapy is typically administered in cycles lasting two to three weeks, for over a period of up to six months [38,155,156]. Due to the high likelihood of mucositis and associated bacterial infections during treatment, there is a high expectation of antimicrobial therapy during each chemotherapy cycle. This prolonged antibiotic use inevitably contributes to the emergence and selective pressure for antibiotic resistance at the end of the treatment [137,157,158]. Therefore, the concurrent use of chemotherapy and antibiotics may significantly contribute to prolonged or even persistent AMR within the gut microbiota of patients.

Additionally, the combined effects of chemotherapy and antibiotics create a multifaceted model that can lead to worse cancer treatment outcomes [21,159]. Firstly, the combination disrupts the commensal gut microbiota, resulting in dysbiosis and facilitating pathogen proliferation [21]. Secondly, it accelerates the activation of the bacterial SOS response, promoting de novo mutations conferring antibiotic resistance [21]. Additionally, this therapy exerts strong selective pressure favoring resistant bacteria within the gut microbiota. Furthermore, it increases the extent of bacterial translocation and systemic infections due to mucositis-related open sores and compromised intestinal barriers [21]. Therefore, while combining chemotherapy and antibiotics may offer therapeutic benefits in cancer treatment, it also has the potential to produce detrimental effects that may outweigh these advantages.

## 7. Strategies to Restore Chemotherapy-Induced Dysbiosis and Mitigate Antibiotic Resistance in CRC Patients

Recent studies have underscored the critical role of the gut microbiota in modulating the efficacy and toxicity of chemotherapy agents, resulting in variability in host responses to treatment [49,59,63,78,160]. However, chemotherapy itself acts as a significant stressor on the gut microbiota, often driving it into an unstable and transient state [24,93]. Notably, chemotherapy is administered in cycles, with each cycle exacerbating gut dysbiosis and mucositis. Moreover, mucositis encourages bacterial translocation, requiring the use of broad-spectrum antibiotics to treat and prevent systemic infections [52,161]. This antibiotic intervention creates selective pressure that drives microbial mutations, diminishes colonization resistance, and promotes the proliferation of resistant pathogens. Consequently, this vicious cycle heightens the risk of infections and accelerates the development of antibiotic resistance, which poses a significant threat to patient survival. Given the heightened vulnerability to infections among such patients, antibiotic resistance could promote CRC-related mortality. This highlights the urgent need for strategies aimed at preventing chemotherapy-induced dysbiosis and the proliferation of antibiotic resistance in CRC patients.

Targeting the gut microbiota has emerged as a promising strategy for managing CRC, aiming not only to mitigate the adverse effects of chemotherapy on the gut microbiota but also to enhance treatment efficacy and improve patients’ quality of life [15,18,162]. Unlike other side effects, the impacts of chemotherapy on the gut microbiota can be ameliorated through dysbiosis restoration [49,162]. Several strategies have been proposed for ameliorating dysbiosis and alleviating associated complications in chemotherapy patients, including fecal microbiota transplantation (FMT), probiotics, and phage therapy [163,164].

Recent studies have demonstrated significant restoration of commensal microbiota and a reduction in antibiotic resistance genes in recurrent *Clostridium difficile* infection patients undergoing FMT [165,166,167]. Although not yet widely available for all diseases, clinical cases indicate that FMT can restore microbial diversity and improve the levels of metabolites such as SCFAs to levels comparable to those of healthy donors [168,169]. In acute myeloid leukemia patients, autologous FMT was shown to restore dysbiosis by increasing proportions of beneficial taxa such as *Ruminococcaceae*, *Clostridiales*, and *Lachnospiraceae*, while decreasing pro-inflammatory taxa from the *Enterococcaceae* and *Enterobacteriaceae* families that predominated during chemotherapy [170]. These findings suggest that FMT may be effective beyond its current FDA-approved application, potentially reversing chemotherapy- and antibiotic-induced gut dysbiosis in CRC patients [169].

Although FMT offers a rapid restoration of gut microbial diversity, probiotic administration ensures a more targeted approach with limited chances of pathogen colonization from donors [171,172,173]. The effectiveness of probiotics can be optimized based on knowledge of key species linked to antibiotic resistance, adverse effects, and specific gut microbiota recovery goals [174,175]. This explains why beneficial outcomes of probiotic use are influenced by factors such as strain selection, dosage, duration of treatment, and host physiology [176]. For instance, a recent study involving nasopharyngeal cancer patients indicated that a probiotics cocktail comprising *Lactobacillus plantarum*, *Bifidobacterium animalis*, *Lactobacillus rhamnosus*, and *Lactobacillus acidophilus* could reduce the severity of chemotherapy-induced mucositis in 21 days by regulating gut dysbiosis and enhancing immune responses [177]. In CRC cases, yeast-based probiotics, particularly *Saccharomyces*, have been reported as adjunctive therapies for CRC [178,179,180]. Furthermore, another study involving CRC patients revealed that the administration of two daily tablets totaling 7 × 10^9^ CFUs *Lactobacillus acidophilus* NCFM and 1.4 × 10^10^ CFUs *Bifidobacterium lactis* Bl-04 improved the gut microbiota by increasing the abundance of butyrate-producing bacteria such as *Faecalibacterium* and *Clostridiales* spp., while decreasing levels of CRC-associated genera like *Fusobacterium* and *Peptostreptococcus* [181]. However, not all probiotic strains are equally effective in restoring chemotherapy-induced dysbiosis; thus, screening for potent strains is essential for developing effective probiotic-based interventions [175,182,183].

In addition to probiotics, prebiotic dietary supplements can serve as substrates to promote the growth of low-abundance commensal bacteria and enhance SCFA production—including acetate, propionate, and butyrate—impacted by chemotherapy treatment [184,185]. A recent study indicated that SCFAs can effectively suppress inflammatory responses induced by 5-FU and maintain the integrity of intestinal mucosal epithelium [186,187]. This could alleviate mucositis and associated effects, thus improving chemotherapy side effects.

Research indicates that dietary habits are crucial in modulating the gut microbial community structure [11,188]. For example, a Western dietary style increases the risk for CRC-associated gut microbiota while CRC patients are recommended to eat a plant-based diet with appropriate dietary fiber intake [189,190,191]. These fibers undergo microbial degradation in the large intestine, increasing essential gut metabolites such as SCFAs [192]. Research also indicates that dietary fiber intake leads to increased abundances of beneficial gut bacteria such as Bacteroidetes and lactic acid bacteria, including species such as *E. rectale*, *Ruminococcus*, and *Roseburia* [193]. Although such interventions may take long to significantly alter the gut microbiota, they remain a viable long-term strategy for ameliorating chemotherapy-induced gut dysbiosis.

Recent microbiome modulation strategies have been proposed, including phage therapy, SCFA supplementation, and bacterial consortium transplantation, with a shared goal to replace opportunistic gut microbes and restore eubiosis [194,195,196]. However, comprehensive studies on their safety, efficacy, and application remain limited. For instance, despite decades of phage therapy use in Eastern Europe, key data on phage amplification, dosing, and timing are still needed to address safety concerns [197]. Therefore, further research with larger study populations is still needed to establish the effectiveness, safety, and optimal treatment duration of these approaches before routine application.

In response to the global public health challenge posed by AMR [198], antimicrobial stewardship is strongly recommended in order to optimize antibiotic use and mitigate the risk of antibiotic resistance among chemotherapy-treated patients, which can either be chemotherapy-induced or arise from prolonged antibiotic use [199,200]. Through these deliberate stewardship efforts, physicians, hospitals, nursing homes, palliative care facilities, and other healthcare providers ensure they prescribe the appropriate antibiotics at the right dose, at the right time, and for the right duration to achieve the best clinical outcomes [201]. Cancer patients stand to benefit significantly from antibiotic stewardship [202], as prior antibiotic exposure is a known risk factor for developing antibiotic-resistant infections [203,204]. A recent study indicates that interventions by antimicrobial stewardship teams (ASTs) have improved clinical and microbiological outcomes, particularly concerning resistant Gram-negative bacteria [205]. These improvements enhance antibiotic susceptibility rates and overall patient outcomes, which is particularly relevant for CRC patients undergoing chemotherapy, given that infections are a major contributor to cancer-related mortality in these patients [206]. Therefore, CRC societies and infectious disease centers should collaborate to guide data collection on variables pertinent to antibiotic resistance. Such collaborations can develop robust antibiotic stewardship and surveillance policies for optimized antibiotic use across all infections [201,207]. For CRC patients, the accurate reporting of deaths attributable to antibiotic-resistant infections should be emphasized in such policies because it could provide a clear understanding of the magnitude of antibiotic resistance [208,209], enabling timely interventions to restrict the spread of resistant infections within oncology hospitals.

## 8. Conclusions

Although chemotherapy remains a cornerstone in the treatment of advanced CRC, its interactions with the gut microbiota highlight a significant aspect of its non-selective toxicity, which extends beyond tumor cells to induce gut dysbiosis. This non-selective toxicity encompasses a range of side effects, including damage to the colorectal epithelium and the disruption of commensal bacteria, resulting in mucositis and the activation of the bacterial SOS response. The compromised gut epithelium, combined with chemotherapy-induced dysbiosis, elevates the risk of pathogenic bacterial overgrowth and bacterial translocation through open sores, leading to severe systemic infections that often require broad-spectrum antibiotics for prevention and treatment. The concurrent use of antimicrobial therapy throughout chemotherapy cycles, along with chemotherapy-induced mutations, creates selective pressure that favors the dominance of antibiotic-resistant bacteria in the gut. This compromises the efficacy of chemotherapy and increases the risk of CRC mortality. Given the critical role of the gut microbiota in influencing both the toxicity and efficacy of chemotherapy, restoring the gut microbial balance is essential. Emerging approaches, such as FMT, probiotics, and prebiotics, present promising strategies for restoring the gut microbiota and mitigating the chemotherapy-induced antibiotic resistance. However, it is important to note that FMT so far has been approved by the FDA for the treatment of recurrent Clostridium difficile infection, implying that clinical studies in other conditions, including CRC, are still needed in order to establish efficacy and safety. Furthermore, not all probiotics are effective in alleviating chemotherapy-induced dysbiosis, underscoring the necessity for screening potent probiotic strains. Therefore, antimicrobial stewardship is recommended in order to optimize antibiotic use and mitigate the emergence and spread of antibiotic resistance among chemotherapy-treated CRC patients. Future research should prioritize reducing the impact of chemotherapy on the gut microbiota and optimize drug–probiotic combinations to improve gut microbiota restoration and chemotherapy outcomes.

## Figures and Tables

**Figure 1 antibiotics-14-00264-f001:**
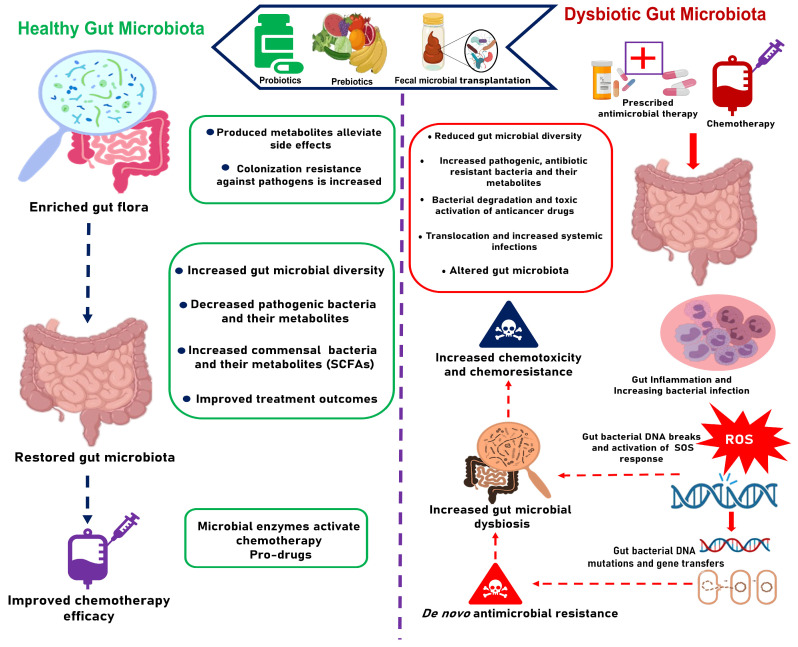
The impact of gut microbiota to the efficacy and toxicity of CRC chemotherapy.

## Data Availability

No new data were created or analyzed in this study. Data sharing is not applicable to this article.

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
