# Peer review of "Exploring the Impact of Chemotherapy on the Emergence of Antibiotic Resistance in the Gut Microbiota of Colorectal Cancer Patients"

_antibiotics, 2025, doi:10.3390/antibiotics14030264_

Round 1
Reviewer 1 Report
Comments and Suggestions for Authors
The manuscript highlights several important aspects regarding antimicrobial resistance in colorectal cancer patients due to both antibiotic therapy and chemotherapy.
Major points
- Lines 68-73 – the new therapeutic options should at least be mentioned here
- Line 109-116 – are these mechanisms of action required for the understanding of the rest of the manuscript? If not, they should be removed
- Figure 1 should be redone to some extent – the left side of the image is somewhat unclear regarding the transition from a healthy gut microbiome to corrected / restored gut microbial dysbiosis
- bacterial names should respect the correct nomenclature – e.g., Escherichia coli; furthermore, after the first time the name appears in the text it should be abbreviated to E. coli etc
- Table 1 – reformat the table; furthermore, some mechanisms are not entirely clear and should be better explained
- Section 1.3 – the effects of different chemotherapy drugs on the microbiome should be summarised in a table
- Section 1.4 – all the highlighted mechanisms of AMR should be summarised in a table
- Line 333-340 – are there data in the literature comparing survival rates or infection incidence between patients receiving antimicrobial prophylaxis and those treated only upon confirmed evidence of infection?
- Line 406 – 420 – specify the probiotic regimens (composition, duration etc) used in the cited studies
- Line 421-429 – although this section initially focused on prebiotics, the discussion transitioned toward probiotics
- Section 1.6 could be further improved by expanding the discussion towards other microbiome modulation strategies
- The numbering of the chapters and sub-chapters needs to be redone
- Some ideas seem to be repeated multiple times throughout the text
Minor points
- “Furthermore, chemotherapy induces sepsis and recurrent septic episodes [31], thus compromising the intestinal barrier and increasing susceptibility to infections [32].” – the logical order would be barrier dysfunction – susceptibility - sepsis
- Line 165-168 – E. coli activates tegafur appears twice
- treatment-limiting diarrhoea – perhaps treatment resistant?
- Line 267 – „evolution of commensal E. coli” – replace evolution
- Line 327 – “hospital-acquired infections, which are a major reservoir for antibiotic-resistant organisms” – infections cannot be a reservoir for resistant bacteria
- Line 341 - survivorship rates
- Line 392 - dysbiosis restoration suggests that these therapies might fully restore the damaged microbiome, which is unlikely
- Line 432 - dual emergence of antibiotic resistance?
Author Response
We are grateful to the reviewer for his/her insightful comments on our manuscript. We have been able to include changes to reflect most of the suggestions provided by the reviewer. Kindly find the detailed responses in the attachment.

Reviewer 2 Report
Comments and Suggestions for Authors
· Lines 37-40 must introduce the aim of the paper and must fit with lines 87-89 please redraft them.
Here the lines: Lines 37-40
This review underscores how prolonged antibiotic use during chemotherapy, combined with chemotherapy-induced bacterial mutations, creates selective pressures that drive de novo antimicrobial resistance (AMR), allowing resistant bacteria to dominate the gut.
Lines 87-89
In this review, we examine how chemotherapy can exacerbate antibiotic resistance in CRC patients through induced dysbiosis or mutations, with implications for patient outcomes
· Line 133 change necessitates for implies
· Figure 1
-separate Corrected Gut of Microbial Dysbiosis because is confuse
- Instead of “Increased colonization resistance against pathogens” use Increased colonization resistance by pathogens
-Instead of “Increased pathogenic and resistant bacteria and their metabolites” use Increased pathogenic, antibiotic resistant bacteria and their metabolites.
-It seems that in red and green charts the size of the text is different in some statements.
-Use Microbial enzymes activate chemotherapy Pro-drugs
-In all charts use appropriately upper and lower case letters
-Table 1 use a type of table without a lot of lines.
-I don´t know if is due to the journal requirements, other wise use titles of the table in bolds.
-Lines 149 -150 use appropriately upper and lower case letters
Lines 127-129 and 154-155 are too redundant.
Lines 165 , 166 Please use name of microorganisms in italics in all the paper!!!!!
Lines 168-170 I disagree with “These findings underscore the potential for gut microbial enzymes and their metabolites to enhance chemotherapy efficacy in CRC. ” since the previous discussion is about the modulation of immune system by producing metabolites such as butyrate. Thus, the roll of the enzymes is not clearly involved. Please modify this statement or move it after line 177.
-Table 1, please try to do not cut the words in the line of Therapy, since is not easy on the eyes.
-Please use appropriately upper- and lower-case letters in the table, titles of each section and all the paper.
-Line 430-431 Please explain what the difference between antimicrobial stewardship and antibiotics use, it seems different, or the use of both terms is a little bit confusing in this paragraph. Please make understandable it.
Please check references since some include doi and other not
Author Response

(The authors gave the same response as above.)

Round 2
Reviewer 1 Report
Comments and Suggestions for Authors
In my opinion, the manuscript has been sufficiently improved to warrant publication. However, some minor aspects would still require addressing.
- Table 2 – specify which studies were done in vitro conditions and which were done in vivo in animal models or patients (the fact that some ATCC strains were used would suggest that at least some of the respective studies were done in vitro)
- Table 1,2,3 – are these all the known interactions between the microbiome and chemotherapy drugs? The quality of the manuscript would be increased if further references were sought and incorporated in these tables
- “This aligns with recent comparative studies indicating that cancer patients undergoing chemotherapy with antimicrobial prophylaxis had a poor survival rate compared to those without prophylaxis [153-156]” – add the respective survival rates in the manuscript
Author Response
We have provided the response in the attachment
